# Impacts of Land Use on Surface Water Quality Using Self-Organizing Map in Middle Region of the Yellow River Basin, China

**DOI:** 10.3390/ijerph191710946

**Published:** 2022-09-02

**Authors:** Liang Pei, Chunhui Wang, Yiping Zuo, Xiaojie Liu, Yanyan Chi

**Affiliations:** 1Institute of Geographic Sciences and Natural Resources Research, Chinese Academy of Sciences, Beijing 100101, China; 2Xinjiang Institute of Ecology and Geography, Chinese Academy of Sciences, Urumqi 830011, China; 3College of Resources and Environment, University of Chinese Academy of Sciences, Beijing 100049, China; 4Foreign Environmental Cooperation Center, Ministry of Ecology and Environment, Beijing 100035, China; 5Chinese Academy of Environmental Planning, Beijing 100102, China

**Keywords:** water quality, land use, self-organizing map, redundancy analysis

## Abstract

The Yellow River is one of the most important water sources in China, and its surrounding land use affected by human activities is an important factor in water quality pollution. To understand the impact of land use types on water quality in the Sanmenxia section of the Yellow River, the water quality index (WQI) was used to evaluate the water quality. A self-organizing map (SOM) was used for clustering analysis of water quality indicators, and the relationship between surface water quality and land use types was further analyzed by redundancy analysis (RDA). The results showed that WQI values ranged from 82.60 to 507.27, and the highest value was the sampling site S3, whose water quality grade was “Likely not suitable for drinking”, mainly polluted by agricultural non-point sources ammonia nitrogen pollution. SOM clustered the sampling sites into 4 groups according to the water quality indicators, the main influencing factors for different groups were analyzed and explored in more depth in relation to land use types, suggesting that surface water quality was significantly connected with the proportion of land use types at the watershed scale in the interpretation of water quality change. The negative impact of cropland on surface water quality was greater than that of other land use types, and vegetation showed a greater positive impact on surface water quality than other land uses. The results provide evidence for water environment conservation based on land use in the watershed.

## 1. Introduction

A healthy water environment is important for ecosystem protection, agricultural and industrial development, residential life, public health, etc. However, surface water is vulnerable to point and non-point source pollution. Therefore, monitoring and analysis of surface water quality are crucial for the sustainable development of the water environment [1,2,3,4]. Types of land use reflect the land use intensities connected with human activities, and it is a key factor affecting water quality [5,6,7,8,9]. In addition, the way that types of land use impact water quality may be spatially different due to the different patterns, intensities, and areas of human activities [10,11]. In the city region, the main land use type is the impermeable surface layer, affecting water quality by increasing the concentration of pollutants such as nutrients and metals in runoff [12,13]. In the countryside region, cropland and vegetation are the main land use types, increasing the concentrations of nitrogen, phosphorous, and pesticide in rivers through agricultural production activities [14,15,16].

The Yellow River is the second longest river in China, which greatly influences the development economy and society and the environmental protection of northern China. Sanmenxia City is located in the middle reaches of the Yellow River, and its topography is complex and diverse, including mountains, hills, river valleys, and plains. The annual rainfall is generally between 400 mm and 700 mm, with more rainfall from June to August, and the annual average temperature is 14.2 °C. It belongs to the semi-arid climate of the continental monsoon type in the warm temperate zone [17]. The major land covers in Sanmenxia City include cropland and impermeable surface layer, and the pollutants produced by different utilization modes enter rivers and affect the water quality [18]. Under the comprehensive function of natural and human factors [19], water pollution and the contradiction between the supply and demand of water resources in the Yellow River Basin became serious in some reaches [20,21,22]. Farmland cultivation and urbanization affected water quality significantly. There is a positive correlation between farmland, urban land, and water pollution [23]. Municipal industrial and residential waste discharges containing organic pollutants, heavy metals, and nutrients can affect water quality. Pollutants from agricultural land affected by fertilizers and pesticides entering water bodies can lead to significant nitrogen and phosphorus pollution in rivers as well as organic pollutants and toxic substances. Forest and grassland play a purifying role in environment protection and river water quality. As a result, it is significant to analyze the relationship between surface water quality and land use types.

Water Quality Index (WQI) can be used to calculate and interpret water quality information, turning complex water quality information into data that can be easily recognized and understood, and the method has been applied to the evaluations of surface water and groundwater in many countries [24,25,26,27,28,29]. The parameters of water quality in the Aksu River were divided into five categories by WQI for further visual processing by GIS tools [25] used the method of WQI to analyze water quality in Mokopane Area, Limpopo, South Africa, and found that all water samplings fell under the poor category. In a word, WQI is considered to be an effective tool for assessing and managing water quality [28].

It is difficult to deal with water quality indicators with linear analysis because of their complexity and variability, while on-linear clustering analysis can explain the complex variations of water quality indicators accurately. Cluster analysis has been proven to be a useful tool to detect surface water quality and water resources by using methods relying on the similarities of water quality parameters [30]. Compared with other cluster analysis methods, the self-organizing map (SOM), a widely known clustering method, has served as a better visual tool to divide the variables ecological and environmental into different groups, and it is widely used in several ways, such as surface water, coastal water, biogeochemical processes, and groundwater assessment [31,32,33,34]. Redundancy analysis (RDA) is an analysis method that can assess the connections of each variable and the relationship between different sets of variables. It is widely used in the studies of environmental factors, including the correlation between natural conditions around sampling sites and the results of the samples. This study was performed with SOM for cluster analysis to study further the main influencing water quality parameter of each cluster, and the RDA was used to analyze the correlation between land use types and water quality at the scale in the watershed.

Therefore, the main purposes of this study are to (1) evaluate the surface water quality in the study area by WQI; (2) classify different water samples by SOM and identify the main influencing four categories; (3) explore the relationship between land use types and water quality indicators with RDA and correlation analysis. The study aims to comprehensively understand the current situation of surface water quality in the middle reaches of the Yellow River, the temporal and spatial variation of water quality, and the impact of land use types on water quality. It will enlighten water environment protection and scientific land management in the Yellow River Basin.

## 2. Materials and Methods

### 2.1. Study Area

The research area is located in Sanmenxia City (33°31′24″–35°05′48″ N, 110°21′42″–112°01′24″ E), Henan Province (Figure 1). The annual rainfall is generally between 400 mm and 700 mm, with more rainfall from June to August, and the annual average temperature is 14.2 °C. It belongs to the semi-arid climate of the continental monsoon type in the warm temperate zone.

### 2.2. Sampling and Selection of Water Quality Indicators

Fourteen water samples were collected once a month from 2016 to 2019. The sampling sites were distributed in the main streams of the Yellow River (S1, S2, S4, S5, S6, S8) and the Yiluo River (S3, S7, S9, S10, S11), which was a tributary of the Yellow River (Figure 1). Sampling sites were set up in accordance with these principles: the locations should be representative of the study area, and surface water (0.5 m) was collected and stored in a preprocessed plastic bottle at each sampling site. All samples were kept in a dark and cold environment until they were taken back to the laboratory for determination. Six water physicochemical indicators were selected, including dissolved oxygen (DO, mg/L), permanganate index (I_Mn,_ mg/L), biochemical oxygen demand (BOD, mg/L), ammonia nitrogen (NH_3_-N, mg/L), total phosphorus (TN, mg/L) and total phosphorus (TP, mg/L). Chemical analysis methods of the indicators used for sampling sites are listed in the Ministry of Ecology and Environment of China (https://www.mee.gov.cn/ywgz/fgbz/bz/ (accessed on 15 August 2021)).

### 2.3. Water Quality Index

The water quality index (WQI) used in the study is regarded as one of the reliable tools for classifying surface water pollution, which integrates the assessment results of different indicators into a general situation of water bodies. Chemical indicators were selected, and the weight of each indicator was labeled from 1 to 4 based on its impact and importance in the overall water quality assessment [35]. Based on previous research [35], the six indicators (DO, I_Mn_, BOD, NH_3_-N, TN, TP) were considered for WQI computation. Referring to previous research results on the use of WQI to evaluate water quality [35,36,37], different weights are assigned in order of the degree of impact on the survival of aquatic organisms. DO with the highest effect on water quality was labeled the weight with 4, TP with the least effect was labeled with a weight of 1, the weights of I_Mn_, BOD, and NH_3_-N were 3, and the weight of TN was assigned a value of 2.

The relative weight of each indicator was calculated as:(1)$$W_{i}=w_{i}/\sum_{i=1}^{n}w_{i}$$
where $W_{i}$ and $w_{i}$ are the relative weight and the weight of each selected indicator, respectively, and n is the number of indicators.

The quality grade of each indicator was calculated as:(2)Qi=CiSi×100
where $Q_{i}$ is the quality grade, $C_{i}$ is the concentration of each indicator, and $S_{i}$ is the standard for each indicator referring to Chinese environmental quality standards for surface water [38]. The WQI model can be defined as:(3)$$\mathrm{WQI}=\sum_{i=1}^{n}W_{i}\times Q_{i}$$

The values of WQI were divided into five levels, and the levels set in this study are shown in Table 1 [21].

### 2.4. Self-Organizing Map

The self-organizing map (SOM), famous for the Kohonen network, was proposed by Teuvo Kohonen in Finland [39]. SOM is an artificial neural network that simulates the characteristics of human brain signal processing and is a high-dimensional visualization algorithm of clustering. There is a clustering function for nonlinear data in the self-organizing map so that it can deal with the high-dimensional data matrix into a two-dimensional or one-dimensional topology [40]. SOM can be divided into two steps: self-organizing training and an output feature map. In self-organizing training, the training sample data undergo competitive learning, and the results are presented on a series of neurons [41,42,43]. The vector inputted of multi-dimensional observation data is connected to each neuron through the weight vector, and the output feature map is related to adjacent neurons through the neighborhood relationship, which can determine the topological structure of the map. The optimal and representative SOM reference vectors are obtained through 4 main training processes, including initialization, competition, selection of winning neurons, and clustering of similar vectors [44].

SOM has been successfully used in the visualization analysis of the quality of both surface water and groundwater. Many studies have successfully analyzed the water quality changes with SOM [44,45]. In this study, the multivariate data matrix of monitoring sites was treated as the input layer of SOM, and the results were presented in the output layer by self-organizing training learning. The SOM Toolbox (http://www.cis.hut.fi/projects/somtoolbox (accessed on 15 August 2021)) of MATLAB was employed for clustering analysis and visualization of water quality monitoring data.

### 2.5. Land Use Data and Redundancy Analysis

The dataset of land use with a resolution radio of 30 m was obtained from the Tsinghua University Database (http://data.ess.tsinghua.edu.cn/ (accessed on 15 August 2021)), and the overall data accuracy was 72.76% [46]. The types of land use were divided into 10 categories: cropland (CR), forest (FR), grassland (GR), shrubland (SR), wetland (WE), water (WA), tundra (TU), impervious surface (IS), bare land (BL) and snow/ice (SI). Considering the actual situation of Sanmenxia City, there is no existence of tundra and snow/ice. Eight major land use types were selected. The land use types of the study area were mainly cropland and forest, including a small part of the impervious surface, and the main land use types along the river were cropland and impervious surface (Figure 2). In order to analyze the impact of land use on the spatial changes in water quality, the proportions of land use in the small watershed were extracted by ArcGIS developed by ESRI. The correlations between the area proportion of each land use type and water quality indexes were analyzed in Origin.

Redundancy analysis (RDA) was adopted to explain the relationships between land use types and water quality indicators. RDA can effectively perform statistical tests on multiple environmental variables and maintain the variance contribution rate of each environmental variable to species variables [47]. In this research, the water quality indicators were regarded as response variables (species variables), and the type of land use in the watershed zone was processed as explanatory variables (environmental variables).

## 3. Results and Discussion

### 3.1. Summary Statistics of Water Quality Parameters

Descriptive statistical analysis of all water quality indicators for each water sampling site from 2016 to 2019 was shown in Table 2, and different water quality indicators showed different CV (coefficient of variance) values. The CV values of NH_3_-N and TP exceed 100%, indicating high variations between sampling sites in both temporal and spatial scales. The average CV values of electrical conductivity (EC), I_Mn_, BOD, and TN were 26.94%, 43.61%, 55.56%, and 43.61%, respectively, showing moderate variability. The pH and DO both showed low variability and were below 10%.

In Figure 3, the changes in water quality during the wet season (from July to October) and dry season (from March to June) were described. Generally, the concentrations of BOD, I_Mn_, NH_3_-N, TN, and TP in the dry season were higher than those in the wet season except for DO, so the water quality in the dry season was worse than that in the wet season. In addition, compared with class III (Chinese environmental quality standards for surface water, GB3838-2002), the content of NH_3_-N and TN exceeded the standard, and TN was the most serious degree, which reflected the heavy nutrient pollution here. The variation of TN and TP concentration sometimes had extreme values at certain sampling sites concentrated in the dry season, while the other samples were generally good. The concentration of TN exceeded the standard rate by 100% at the five nationally controlled sections in the upper Yellow River, and the maximum values could reach 2.41, 2.82, 3.26, 2.91, and 3.94 times the standard limit, respectively, which indicated that the water quality in the midstream area is influenced to some extent by the water coming from upstream [22]. However, in the lower reaches of the Yellow River, TN is 1% above the national standard [48]. It can be concluded that water quality varied considerably from region to region and from time to time.

### 3.2. Spatial-Temporal Distribution Characteristics of Water Quality

The WQI results of water samples are shown in Figure 4. Significant differences in the spatial distribution of WQI indicated significant differences in water quality in the study area (Figure 4a). The WQI values ranged from 82.60 to 507.27, with an average value of 231.17, and the monitoring site S3 had the highest WQI value. According to the classification grade (Table 1), the water quality in the study area included 4 categories: good, poor, very poor, and likely not suitable for drinking [21]. The water quality of S3 was “Likely not suitable for drinking “, which was the worst among all sampling sites. The main pollutant in S3 was NH_3_-N, which is mainly polluted by human activities (Figure 2). In addition, S3 is located in the upstream area of the river, where river flows are low. Many livestock farms lead to the discharge of large amounts of pollutants into water bodies, resulting in high pollutant concentrations, with NH_3_-N levels significantly exceeding [49].

Additionally, only 14% of the water samples were good level and suitable for daily drinking. Approximately 65% of the water samples were poor and not suitable for daily drinking, 14% were very poor, very poor, and 7% were divided into “likely not suitable for drinking”. According to WQI, the water samples unsuitable for drinking water were mainly concentrated in the northeast, and the main land use types of small watersheds in this region were cropland and impervious surface. This result reflected those agricultural activities and urban development negatively affected nearby water quality [21]. WQI results showed (Figure 4b) that WQI values gradually decreased as time went by, indicating that water quality was gradually improving. Additionally, the concentration of pollutants (I _Mn_, BOD, NH_3_-N, TN, and TP) in the river has been reduced by 32.4%, 45.6%, 59.8%, 2.3%, and 45.8%, respectively. The water quality in the middle reaches of the Yellow River has become better due to the government’s attention to the erosion problem, the reduction of point and non-point source pollutant discharges along the river, and the continuous promotion of the work of returning farmland to the forest (the area reached 2,049,000 mu) (http://www.henan.gov.cn/ (accessed on 15 August 2021)).

### 3.3. Indicators of Each Group Based on Self-Organizing Map

The results of SOM were shown in Figure 5, and the total monitoring sites were clustered into four groups using the SOM [6]. The size of the map was 98 (7 × 14), with the minimum values of QE and TE, 0.432 and 0.000, respectively. The unified distance matrix and characteristic plane evaluated the similarity of different pollution indicators. NH_3_-N, TN, and TP are similar in spatial distribution, with the highest values at the bottom-left and the lowest at the top-right part of the results, indicating the combined pollution of these pollutants in the corresponding areas. I_Mn_ and BOD had similar spatial distributions, with the highest values at the bottom and the lowest values at the top. The highest value of DO was on the left of the panel, and the lowest value was on the right.

Cluster A (S2, S7, S12, S3) significantly differs from other clusters. The overall value of WQI was between 50 and 100, and the water quality condition was good, mainly because the overall impact factors of water quality were relatively low. Additionally, the type of land use in cluster A was mainly vegetation, which improved water quality. Cluster B (S3, S4, S10, S14) was distributed in the upper right corner of the panel, and the water quality condition was poor. With the shadow analysis of water quality, DO was the main influencing factor. Cluster C (S5, S9) and cluster D (S1, S6, S8, S11) had relatively poor water, with WQI values between 100 and 200. The types of land use around clusters C and D were residential areas and agriculture, with relatively high concentrations of pollutants due to human activities, resulting in poor water quality.

### 3.4. Impacts of Land Use on Water Quality

Generally, the types of land use were divided into 5–8 categories, and more detailed separation would not change the results of the impact on water quality [19]. As some of the land use types in the watershed are relatively small in size, the relevant land use types have been classified into one category for better analysis. Therefore, the land use types were reclassified into four categories: agriculture (Agri), which included cropland, settlement (Sett), which included impervious surface and bare land, vegetation (Vege), which included forest, water (water and wetland), grassland and shrubland, respectively. Correspondingly, the main land use types in the study region were cropland, forest, and grassland, and vegetation was mainly distributed in the central and northern of this area. The correlation analysis results between indicators of water quality and types of land use in Figure 6a showed spatial variability in the relationship in different regions. It can be learned that DO had a positive correlation with Vege and a negative correlation with Agri, with a correlation coefficient of 0.56 and −0.58, respectively. I_Mn_, BOD, NH_3_-N, TN, and TP were positively correlated with Agri with the correlation coefficients of 0.53, 0.55, 0.47, 0.55, and 0.75, respectively. However, these indicators were negatively correlated with Vege and water, which suggested that the increasing agricultural land would increase the concentration of pollutants in water and lead to a decrease in DO in water [50].

On the contrary, increasing vegetation area and water area would reduce the concentration of pollutants in water and increase the DO in water. The results displayed that agricultural land had a negative effect on water quality, while vegetation and water area had a positive effect on water quality. The positive impact was because of the interception and absorption of pollutants by nearby natural vegetation [51]. There was no obvious correlation between impervious surface and water quality indicators in correlation analysis.

The method of RDA further explored the connections between land use types and water quality indicators. In Figure 6b, the arrows represented environmental factors, and the length of the lines indicated the level of correlation between environmental factors and sample indicators. The included angle between different arrow lines indicated the correlation, and the acute angle indicated a positive correlation. The results of RDA showed a significant negative correlation between TP, BL, and GR, a positive correlation between it and IS and CR, and DO have a significant negative correlation with IS and CR, and a positive correlation between BL and GR. BOD had a positive correlation with I_Mn_, which is significant, while the positive correlation between NH_3_-N and TN was not significant. The land use types of the sampling sites could well explain the differences in TN and TP concentrations, similar to the results of Zhou [51] and Shen [52]. The results further proved that TN, TP, CR, and IS were significantly correlated, and the explanatory degree reached 95.16%, which could meet the accuracy requirements.

## 4. Conclusions

The water quality and spatio-temporal variations of six indicators of the Sanmenxia section in the Yellow River Basin from 2016 to 2019 were chosen and discussed. The WQI method, SOM method, and RDA method were used to comprehensively evaluate the quality of surface water. The research showed that the water quality in the middle region of the Yellow River Basin was good and gradually improved from 2016 to 2019. The relationship between land use types and water quality was further discussed using redundancy analysis and a self-organizing map. Agriculture had a negative effect, but vegetation had a positive effect on surface water quality. The relationship between land use types and water quality in the middle reaches of the Yellow River is further analyzed and clarified at the spatial scale in this study. However, there may be differences in different study areas and even differences in time and space scales within the same study area the interaction mechanism between types of land use and surface water quality was complex, and further study is needed. The improvement of water quality may be related to the progress of environmental management and the residents’ awareness of environmental protection. Therefore, the government should pay further attention to the treatment and improvement of the water quality of the Yellow River, formulate relevant policies to reduce the discharge of pollutants (such as domestic sewage, agricultural pollution, and other sources) along the river, and continuously promote measures to return farmland to forests, to improve the water quality of the Yellow River basin.

## Figures and Tables

**Figure 1 ijerph-19-10946-f001:**
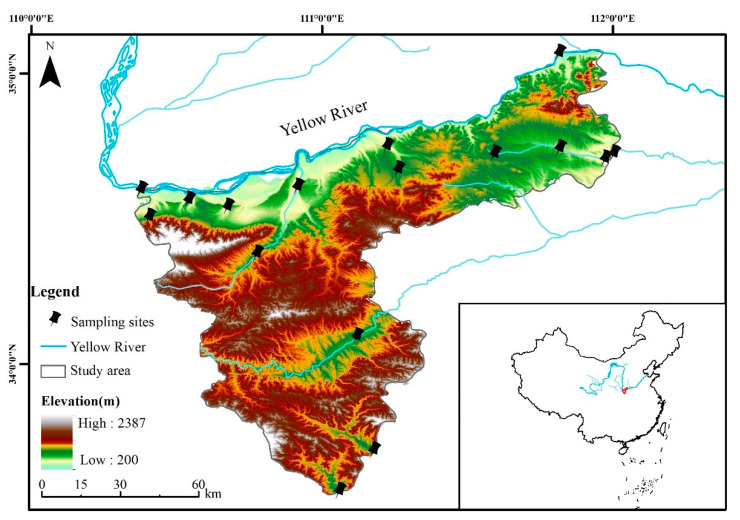
Location of the study area and sampling sites.

**Figure 2 ijerph-19-10946-f002:**
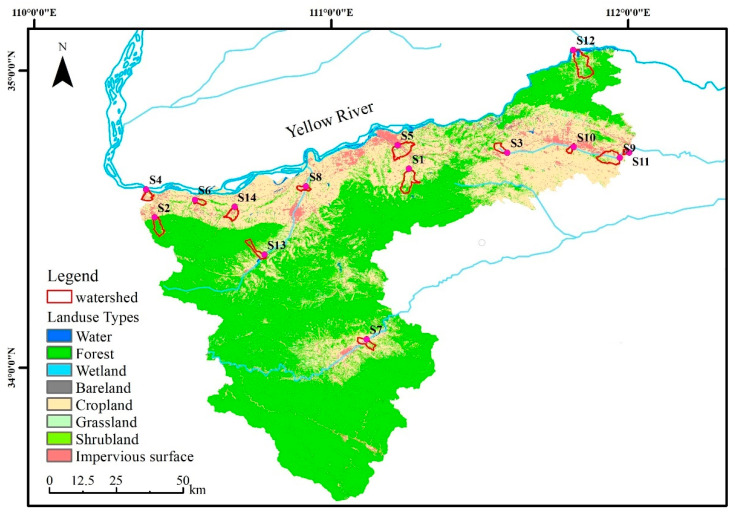
The land use types of the study area.

**Figure 3 ijerph-19-10946-f003:**
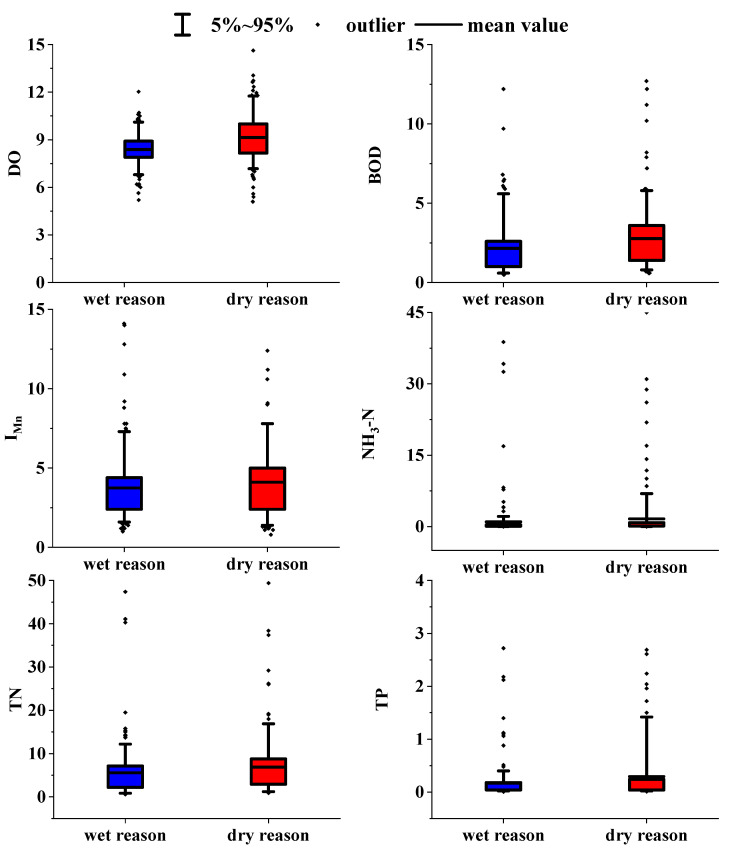
Box plot of the water quality in the wet season (from July to October) and dry season (From March to June).

**Figure 4 ijerph-19-10946-f004:**
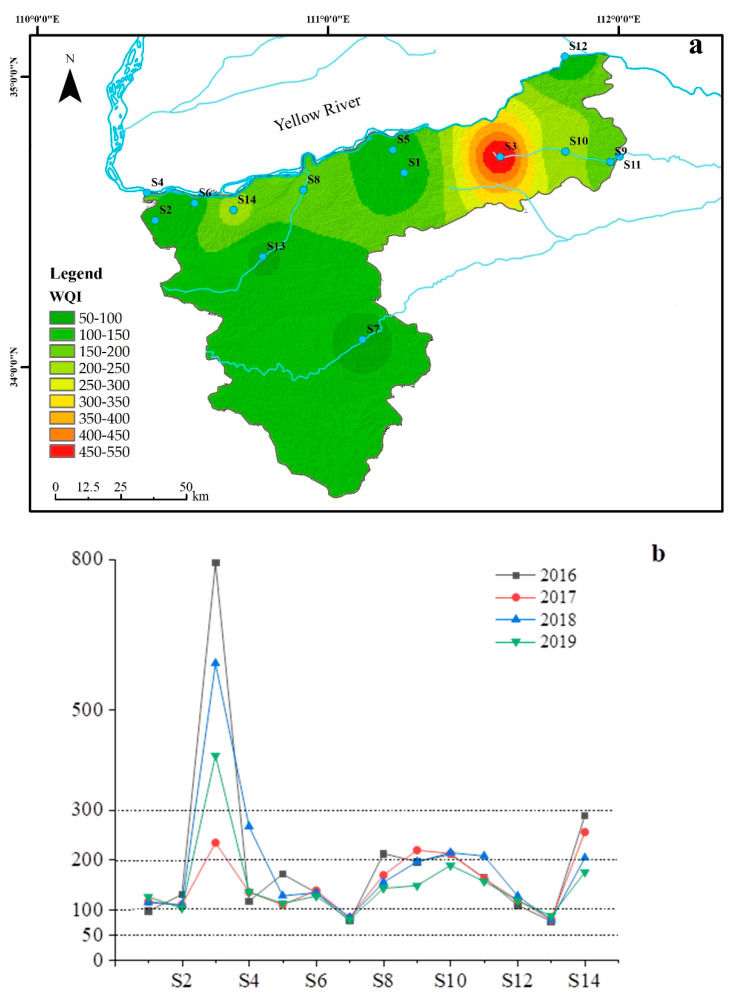
Spatial distribution of WQI values and variation in WQI values of sampling sites from 2016 to 2019. (**a**) distribution of WQI values; (**b**) Range variation of WQI values.

**Figure 5 ijerph-19-10946-f005:**
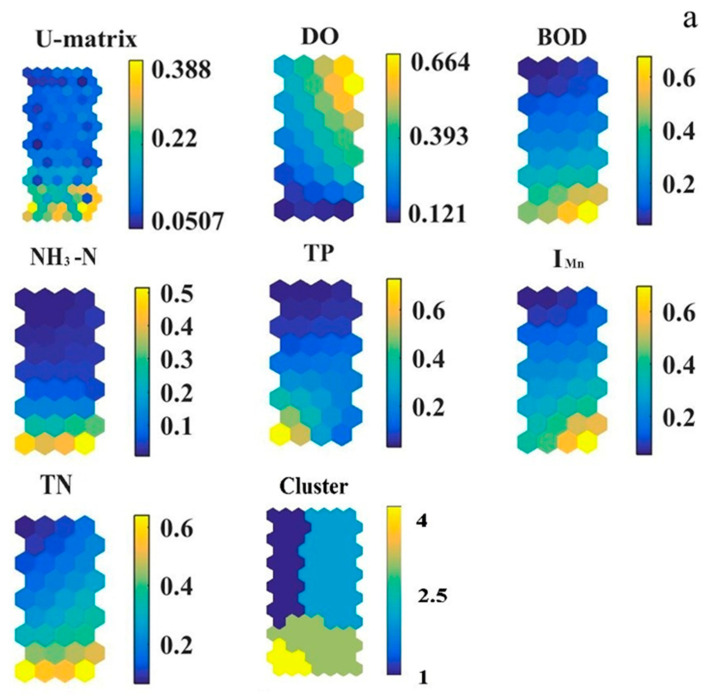
(**a**) Visualization of six water quality indicators for all sampling periods and sites by SOM Toolbox, (**b**) classification of sampling sites.

**Figure 6 ijerph-19-10946-f006:**
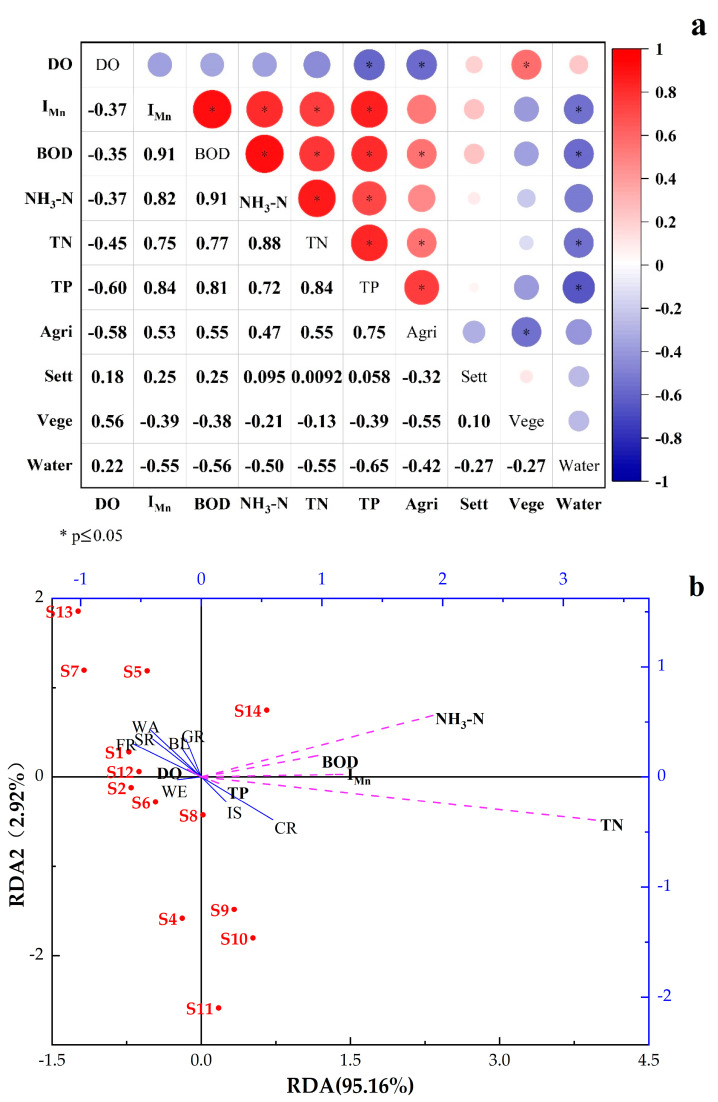
(**a**) Correlation analysis results and the relationships between land use types and water quality indicators, (**b**) redundancy analysis of the relationships between land use types and water quality indicators.

**Table 1 ijerph-19-10946-t001:** Classification of the water quality index (WQI).

Ranking	Water Quality
<50	Excellent
50–100	Good
100–200	Poor
200–300	Very poor
>300	Likely not suitable for drinking

**Table 2 ijerph-19-10946-t002:** Descriptive statistics of water quality indicators of the study area.

Time	Parameter	pH	EC	DO	I_Mn_	BOD	NH_3_-N	TN	TP
2016	Mean	8.07	77.03	8.79	4.79	2.94	2.04	6.02	0.24
	SD	0.14	12.95	0.57	2.57	2.00	4.05	4.75	0.27
	Range	7.83–8.27	54.60–96.60	8.00–9.80	1.80–11.80	0.90–8.60	0.10–15.59	1.74–21.00	0.04–1.09
	CV (%)	1.77	16.80	6.48	53.66	68.02	198.11	78.84	112.43
2017	Mean	7.91	74.05	9.16	3.84	2.54	1.17	5.72	0.17
	SD	0.15	19.72	0.49	1.75	1.37	1.74	3.11	0.17
	Range	7.63–8.21	43.60–115.10	8.40–9.90	1.50–8.00	1.00–5.70	0.14–6.50	1.51–13.40	0.04–0.68
	CV (%)	1.96	26.63	5.39	45.59	54.15	148.30	54.41	103.57
2018	Mean	7.92	77.85	9.53	3.24	2.03	1.32	6.64	0.15
	SD	0.22	21.66	0.37	1.18	1.09	3.17	4.21	0.20
	Range	7.52–8.19	46.00–126.20	8.80–10.00	1.70–5.40	0.80–4.90	0.07–12.25	1.37–17.60	0.02–0.82
	CV (%)	2.75	27.82	3.86	36.44	53.84	241.33	63.43	136.81
2019	Mean	7.86	85.47	9.37	3.24	1.96	0.82	5.88	0.13
	SD	0.15	31.22	0.61	1.26	0.90	1.79	3.70	0.17
	Range	7.57–8.15	42.40–142.40	8.10–10.20	1.60–6.50	0.90–4.30	0.09–6.83	1.55–16.30	0.03–0.66
	CV (%)	1.90	36.53	6.53	38.76	46.21	218.36	62.82	123.81

Notes: Mean: arithmetic mean; SD: standard deviation; CV: coefficient of variance.

## Data Availability

The datasets used and/or analyzed during the current study are avail-able from the corresponding author on reasonable request.

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
