# Peer review of "Impacts of Land Use on Surface Water Quality Using Self-Organizing Map in Middle Region of the Yellow River Basin, China"

_ijerph, 2022, doi:10.3390/ijerph191710946_

Round 1

Reviewer 1 Report

The study aims to comprehensively understand the current situation of surface water quality in the middle reaches of the Yellow River, the temporal and spatial variation of water quality and the impact of land use types on water quality. I recommend a major revision and there might be some questions for this article to be further clarified.

The specific comments are as follows:

1.        In the introduction, there is need to provide sufficient background information so that readers can easily understand. Authors should also encapsulate why this research needs to be conducted. Clearly show what research works have been done before by other authors, discuss what they have done and justify your research by showing gaps. Is there a gap in knowledge that work on this topic could help to fill or a controversy that it might help to resolve? Please revise the introduction.

2.        The title of Table 1 is confusing, please modify it.

3.        Line 188, please explain the meaning of CV in the text.

4.        Line 193, not "In fig 2", but figure 3. Meanwhile, it is not recommended to express the wet season and dry season as high and low in the figure.

5.        Line 198, please check GB3838-83-2002.

6.        In the Results and Discussion section, please add some discussions.

7.        Please use a uniform mapping format, as in Figure 5, with uniform a, and b serial number placement.

8.        Line 285, "General speaking,  the longer line means greater relationship". Please find the references supporting this sentence and check this part. This statement should be problematic.

9.        The axis titles in Figure 6 should be RDA1 and RDA2.

10.    Please revise the conclusion section. The authors need to summarize the key findings and recommendable measures based on the findings of the study.

Author Response

We would like to thank you for the careful, constructive and helpful comments and suggestions, which improved the manuscript. We revised the manuscript accordingly, and detailed corrections are listed below.

Reviewer 2 Report

This study aims to clarify the impact of land use types on water quality in the Sanmenxia section of the Yellow River. The water quality index (WQI) and self-organizing map (SOM) were used to evaluate the water quality and their classification, and the relationship between surface water quality and land use types was further analyzed by redundancy analysis (RDA). I suggest a major revision of this manuscript based on the following reasons.

1. Lines 20-21, please add the dominant results of SOM. The sample sites were classified into 4 groups is not the key result of this method.

2. The study area is the Sanmenxia section of the Yellow River. Please add detailed information on the Sanmenxia section of the Yellow River in the Introduction.

3. Lines 121-124, why do you think DO is the most important factor affecting the water quality in this study? What are the references or reasons for the orders of the importance of these factors?

4. Lines 167-168, “eight major land use types were given different weights because of the different impacts on water quality from human activities.” This sentence is vague.

5. Lines 214-216, “The main pollutant in S3 was NH3-N, which was mainly polluted by agricultural non-point sources from cropland (Fig. 2).” This conclusion is not creditable. The major land use types of many sample sites were cropland. Why was the water quality in S3 so serious?

6. Lines 226-229, please give more discussion and detailed information about the conclusion of the water quality trend in this study.

7. Lines 263-266, what is the relationship between reclassified land use types and the spatial distribution of land use types in the study? I think the sentence in Lines 264-266 (Correspondingly, main types …… area) was unnecessary.

Author Response

(The authors gave the same response as above.)

Reviewer 3 Report

In this study, analyze the impact of land use types on water quality in the Sanmenxia section of the Yellow River, the water quality index (WQI) was used to evaluate the water quality. Additionally, this paper is relatively well-organized. I recommend a minor revision and there might be some questions for this article to be further clarified.

The specific comments are as follows:

1. In the lines 53-54, please add some previous studies about water quality and land use types to support your ideas which was “ A a result, it is significant to analyze the relationship between surface water quality and land use types.”

2. Figure 1/2/4/5 lacks geographic information and add the coordinate of latitude and longitude.

3. In the line 93, “ Fig. 2” should be changed to “Fig. 3”. Please confirm it.

4. In the line 295, “General speaking” should be changed to “generally speaking”. Please check for other grammar problems.

5. The conclusion should be more concise. Authors need to present a summary of key findings and suggestible measures based on the study result.

Author Response

(The authors gave the same response as above.)

Round 2

Reviewer 1 Report

The authors have addressed all my comments